# CT-Determined Sarcopenia in GLIM-Defined Malnutrition and Prediction of 6-Month Mortality in Cancer Inpatients

**DOI:** 10.3390/nu13082647

**Published:** 2021-07-30

**Authors:** Francisco José Sánchez-Torralvo, Ignacio Ruiz-García, Victoria Contreras-Bolívar, Inmaculada González-Almendros, María Ruiz-Vico, Jose Abuín-Fernández, Manuel Barrios, Emilio Alba, Gabriel Olveira

**Affiliations:** 1Unidad de Gestión Clínica de Endocrinología y Nutrición, IBIMA, Hospital Regional Universitario de Málaga, Universidad de Málaga, 29007 Málaga, Spain; fransancheztorralvo@gmail.com (F.J.S.-T.); jose.abuin.fdez@gmail.com (J.A.-F.); 2UGG Endocrinology and Nutrition at the University Hospital San Cecilio, 18000 Granada, Spain; victoriaconbol@gmail.com; 3Unidad de Gestión Clínica de Radiodiagnóstico, Hospital Regional Universitario de Málaga, 29010 Málaga, Spain; inma.almendros@gmail.com; 4Unidad de Gestión Clínica de Oncología Médica, Hospital Regional Universitario de Málaga, Universidad de Málaga, 29010 Málaga, Spain; maria.ruvic@gmail.com (M.R.-V.); ealbac@uma.es (E.A.); 5Unidad de Gestión Clínica de Hematología y Hemoterapia, Hospital Regional Universitario de Málaga, 29010 Málaga, Spain; mabagar@gmail.com

**Keywords:** computed tomography, GLIM criteria, malnutrition, oncology, body composition

## Abstract

Our objective was to evaluate the clinical application of third lumbar vertebra (L3)-computer tomography (CT)-determined sarcopenia as a marker of muscle mass in cancer inpatients diagnosed with malnutrition according to the Global Leadership Initiative on Malnutrition (GLIM) criteria and to establish its association with 6-month mortality. Methods: This was an observational, prospective study in patients from an inpatient oncology unit. We performed a nutritional assessment according to GLIM criteria, including muscle cross-sectional area at L3 by CT and skeletal muscle index (SMI). Six-month mortality was evaluated. Results: A total of 208 patients were included. The skeletal muscle cross-sectional area at L3 was 136.2 ± 32.5 cm^2^ in men and 98.1 ± 21.2 cm^2^ in women. The SMI was 47.4 ± 12.3 cm^2^/m^2^ in men and 38.7 ± 8.3 cm^2^/m^2^ in women. Sarcopenia (low SMI) was detected in 59.6% of the subjects. Using SMI as a marker of low muscle mass in application of GLIM criteria, we found 183 (87.9%) malnourished patients. There were 104 deaths (50%) at 6 months. The deceased patients had a lower skeletal muscle cross-sectional area (112.9 ± 27.9 vs. 126.1 ± 37.8 cm^2^; *p* = 0.003) and a lower SMI (41.3 ± 9.5 vs. 45.7 ± 12.9 cm^2^/m^2^; *p* = 0.006). An increased risk of 6-month mortality was found in malnourished patients according to GLIM criteria using SMI (HR 2.47; 95% confidence interval 1.07–5.68; *p* = 0.033). Conclusions: Low muscle mass, assessed by L3-CT, was observed to affect more than half of cancer inpatients. The deceased patients at 6 months had a lower skeletal muscle cross-sectional area and SMI. Malnutrition according to GLIM criteria using CT-determined sarcopenia was shown to adequately predict 6-month mortality.

## 1. Introduction

Sarcopenia is defined by the European Working Group on Sarcopenia in Older People as “a progressive and generalized skeletal muscle disorder that is associated with increased likelihood of adverse outcomes including falls, fractures, physical disability and mortality” [1,2]. It is estimated that between 38% and 70% of patients with cancer suffer from sarcopenia [3]. Healthy adults older than 40 years lose around 1–1.4% of muscle mass per year [4], and in patients with cancer this rate could be up to 24 times higher [5,6]. It has been demonstrated that cancer and its treatment accelerate muscle mass loss and, furthermore, that patients lose muscle mass continuously during treatment [5,7,8].

The etiology of low muscle mass in patients with cancer is multifactorial, being caused mainly by a negative energy balance produced by a decrease in protein synthesis and an increase in protein degradation [9].

Sarcopenia is associated with asthenia, fatigue, worse physical function, and a reduced survival rate [10,11,12,13]. Most studies show a significant decrease in general survival in patients with sarcopenia compared to patients without this condition and independently of the stage or location of the primary cancer [3].

The Global Leadership Initiative on Malnutrition (GLIM) criteria were published in 2018 and aimed at building a global consensus on core diagnostic criteria for diagnosing malnutrition in adults in clinical settings [14]. To detect malnutrition, these guidelines recommend a nutritional assessment considering phenotypic and etiologic criteria. Reduced muscle mass is the phenotypic criterion most difficult to define owing to the existence of diverse methods that can be used for its determination. Thus, the consensus proposes several techniques, including dual-energy X-ray absorptiometry, bioelectrical impedance analysis, computed tomography, or magnetic resonance imaging. 

Computed tomography (CT) is considered to be a highly precise tool for assessing body composition because it can distinguish different tissues [15,16,17]. Additionally, it shows excellent inter-observer reliability [18]. Nevertheless, performing a CT only in order to evaluate body composition has the potential complications associated with the radiation involved [18]. Still, the use of CT to evaluate body composition has become an adequate approach in oncology since many patients undergo it for diagnosis, staging, or follow-up purposes. This is of great relevance, as an early identification of risk factors like sarcopenia and sarcopenic obesity allows for early dietary and physical exercise interventions. 

The third lumbar vertebra (L3) level on CT has been validated as the standard reference point for evaluating body composition in patients with cancer. This technique has been used to assess muscle mass in subjects with and without neoplasm [19,20,21,22] and adequately predicts total muscle mass [21,23]. There are studies that have established cut-off points for diagnosing sarcopenia by means of the skeletal muscle index (SMI) and that have found significant associations between this condition and an increased mortality in patients with cancer [24,25].

After the launch of the GLIM consensus, it has become important to validate their relevance for the clinical practice. Currently, most of the tools that assess nutritional status are based on their ability to predict clinical outcomes [26,27,28]. Up until now, there have been some studies that have evaluated if the GLIM criteria can be used to predict mortality in patients with cancer [29,30,31]. However, as far as we know, studies assessing the use of CT to measure muscle mass in the application of GLIM criteria as a marker of adverse clinical outcomes are scarce [32,33]. Moreover, we are unaware of any study that has been carried out in hospitalized patients with cancer. The prevalence of cancer-associated malnutrition is one of the highest among all inpatient groups, which highlights the importance of its assessment in this population [34,35,36]. 

Our objective was to evaluate the clinical application of L3-CT-determined sarcopenia as a marker of muscle mass in cancer inpatients diagnosed with malnutrition according to GLIM criteria and to establish its association with 6-month mortality.

Our hypothesis is that L3-CT-determined SMI is a good tool to evaluate muscle mass in cancer inpatients and its use, applying GLIM criteria, adequately predicts 6-month mortality.

## 2. Materials and Methods

This was an observational, prospective study that was conducted from October 2017 until April 2018 and included the patients admitted to the oncology ward at the Hospital Regional Universitario de Málaga. The inclusion criteria included the presence of an active cancer, an estimated length of stay greater than 48 h, and signing the informed consent. The exclusion criteria included a readmission before 30 days and a situation of actively dying. We selected the patients that had undergone an abdomen and pelvis CT during the 3 months preceding the hospital admission and during the assessment performed at admission (Figure 1). We collected data about the type of neoplasm, tumor stage, and cause of admission.

### 2.1. Assessment of the Nutritional Status: Malnutrition According to GLIM Criteria

We performed a nutritional assessment within the first 24 h after admission, which was carried out according to GLIM criteria. 

For achieving a diagnosis of malnutrition, the presence of at least one phenotypic criterion and one etiologic criterion is required [14].

For that purpose, anthropometric measures were obtained. Weight was assessed with a weighing scale (SECA 665, Hamburg, Germany), and height was obtained by a stadiometer (Holtain Ltd., Crymych, UK). With these two measurements, body mass index (BMI) was calculated.

We evaluated the following as phenotypic criteria: more than 5% of unintentional weight loss in 6 months, a low BMI (BMI below 20 kg/m^2^ for age < 70 years; and BMI below 22 kg/m^2^ for age ≥ 70 years), and/or decrease in muscle mass determined by CT.

The evaluation of etiologic criteria included a reduced intake or assimilation and/or the presence of disease burden or inflammatory condition (assessed by the Glasgow prognostic score, a cumulative inflammation-based cancer-prognostic marker) [14,37]. 

### 2.2. Body Composition Analysis by Computed Tomography

Triphasic abdominal CTs were analyzed for a range of body composition measures using electronically stored CT images, which had been routinely taken for diagnostic purposes. The third lumbar vertebra (L3) was used as a standard landmark to measure muscle cross-sectional area. Skeletal muscles were identified via Hounsfield unit thresholds (−29 to +150). The total cross-sectional area of skeletal muscles at L3 was computed by OsiriX software (v9.5, Pixmeo, Geneva, Switzerland), as previously described [23]. Skeletal muscle index was calculated (cm^2^/m^2^) to normalize L3 skeletal muscle cross-sectional area for stature. Muscle and adipose tissue radiation attenuation were also determined.

Based on the L3 SMI, sarcopenia was defined using Martin’s sex-and-BMI-specific cut-off points, devised from a cancer population: 43 cm^2^/m^2^ for males with BMI < 25.0 kg/m^2^ and 53 cm^2^/m^2^ for males with BMI > 25.0 kg/m^2^; the cut-off point for sarcopenia in females was 41 cm^2^/m^2^, irrespective of BMI). Low muscle attenuation was defined as <41 Hounsfield units (HU) in patients with BMI < 25.0 kg m^2^ and <33 HU in patients with BMI > 25.0 kg/m^2^ [25]. 

### 2.3. Other Body Composition Techniques

Measurement of triceps skinfold was performed using a Holtain constant pressure caliper (Holtain Ltd., Crymych, UK) in the dominant limb, and the mean was calculated. Fat free mass (FFM) and fat free mass index (FFMI) was estimated according to the formulas of Siri and Durnin [38,39].

BIA was performed with Akern BIA-101/Nutrilab analyser (Akern SRL, Pontassieve, Florence, Italy). The measurements were performed in the supine position, with abducted upper (30°) and lower (45°) limbs. Dedicated software (Bodygram Dashboard from AKERN, Pontassieve, Florence, Italy) was used to perform analyses of the results.

### 2.4. Data Analysis 

Statistical analyses were performed using IBM SPSS Statistics 22.0 (SPSS Inc., Chicago, IL, USA, 2013). Quantitative variables were presented as the mean ± standard deviation. Large sample theory [40,41] was employed to use parametric tests due to our sample size. Differences between paired observations were determined using the student paired *t*-test. Comparison between qualitative variables was done with the Chi-square test. Association between body composition measurements was analyzed using the Pearson’s correlation coefficient. A multivariate Cox regression model was used to estimate the hazard ratio (HR) of mortality at six months in malnourished patients, after controlling for age, sex, and tumor stage. Survival curve was evaluated using the Kaplan–Meier method. For calculations, the significance was set at *p* < 0.05 for two-tailed tests. 

## 3. Results

After applying the inclusion and exclusion criteria, a total of 208 patients were evaluated (Figure 1). Their mean age was 60.5 ± 12.9 years and 55.3% were male. Their general characteristics are shown in Table 1.

Most of the patients (197, 94.7%) had an advanced disease (20.7% stage III, 74% stage IV), with lung cancer being the most frequent type of neoplasm (59 patients, 28.3%), followed by breast cancer (31 patients, 14.9%), esophagogastric cancer (27 patients, 13%), and colon cancer (25 patients, 12%). 

Mean BMI was 24.66 ± 5.07 kg/m^2^. Almost half of the patients (90, 43.5%) were overweight or obese. 

Table 2 shows the body composition of the patients determined by CT, anthropometry, and bioelectrical impedance analysis.

The correlation between muscle parameters derived from L3-CT and other techniques for assessing body composition is shown in Table 3.

Sarcopenia (defined by low SMI) was detected in 59.6% of the patients—64 women (68.8%) and 60 men (52.2%). From the women with low SMI, 17 (18.3%) had a BMI ≥ 27 kg/m^2^ and were considered as sarcopenic obese. From men, 13 (11.3%) had a BMI ≥ 27 kg/m^2^ and were also considered as sarcopenic obese. Low muscle attenuation (myosteatosis) was present in 136 (65.4%) patients.

After six months, 50% (104) of the patients were deceased. Table 4 shows a comparison between these patients and survivors’ CT-determined body composition. 

Through the application of GLIM criteria for malnutrition, we detected low BMI in 20.2% (42) of the patients, with 146 (70.9%) of them presenting unintentional weight loss (more than 5% in 6 months). In addition, 95.6% (199) of patients had a Glasgow prognostic score above 0, presenting disease-related inflammation.

Ultimately, using SMI from L3-CT as a marker of low muscle mass, we found 183 (87.9%) malnourished patients.

According to Kaplan–Meier survival analysis, malnourished patients had significantly lower survival (Figure 2). Malnourished patients according to GLIM criteria using SMI thresholds for sarcopenia presented an increased risk of 6-month mortality after adjusting for sex, tumor stage, and age (HR 2.47; 95% confidence interval 1.07–5.68; *p* = 0.033).

## 4. Discussion

This study aimed to evaluate the use of L3-CT as a determinant of muscle mass in the application of the GLIM criteria for the diagnosis of malnutrition. Our main finding was that L3-CT-determined sarcopenia is an appropriate marker of low muscle mass in the application of GLIM criteria in cancer inpatients, and adequately predicts 6-month mortality.

The prevalence of malnutrition in our study, according to GLIM criteria and using SMI in CT, was high (87%); this figure is similar to the one previously found with other techniques to assess muscle mass [29]. 

Despite the high prevalence of malnutrition found in our series, only 20.2% of subjects presented a low BMI [14,42] and there was a moderate prevalence of sarcopenic obesity. This fact could indicate that the SMI is capable of detecting malnutrition even in patients with sarcopenic obesity. The presence of both sarcopenia and obesity leads to particularly poor clinical outcomes. This may be related to the combination of these two conditions, or to the deficient detection of sarcopenia in cases in which the excess of adipose tissue conceals the loss of muscle mass. It should be noted that sarcopenic obesity has been specifically associated with a lower survival rate in several populations [24,43].

In our population, both SMI and muscle mass measured by L3-CT were lower in patients that died at 6 months from hospital admission. Huang et al. demonstrated that a low SMI (defined according to a previous study performed by the same authors) was associated with a lower survival rate in the long term after undergoing a total gastrectomy due to gastric cancer in the elderly. Nonetheless, only GLIM-defined malnutrition (using SMI for assessing muscle mass) remained as an independent risk factor after adjusting for TNM stage and other covariables [32]. 

In our study, GLIM criteria using SMI as a marker of low muscle mass adequately predicted 6-month mortality. The relationship between GLIM-defined malnutrition and survival rate in patients with cancer has been evaluated in several previous studies. In a preceding study by our group, the application of GLIM criteria using hand grip strength or fat-free mass index through anthropometry to assess muscle mass adequately predicted 6-month mortality in cancer inpatients [29]. Zhang et al. showed that GLIM-defined malnutrition predicts general survival in advanced-age patients with cancer, using calf circumference measurement to evaluate muscle mass [30]. De Groot et al. demonstrated that GLIM-defined malnutrition was associated with 1-year mortality in patients with cancer. In this study, hand dynamometry was used as a muscle mass marker, but this did not improve the prediction of malnutrition or mortality compared to the use of GLIM criteria without considering the presence of reduced muscle mass [31].

Another study by Huang et al. [33] evaluated the predictive value of GLIM criteria, using SMI as a marker of muscle mass, for diverse clinical outcomes in overweight patients who underwent a total gastrectomy due to gastric cancer. They found that the criteria failed to predict postoperative complications, but they did succeed in predicting general and disease-free survival [33].

There are multiple techniques to determine low muscle mass in the application of the GLIM criteria, so its correlation capacity is especially relevant. In our study, we found that the correlation between the muscle parameters determined by L3-CT and other muscle mass parameters (specifically, anthropometry and BIA) is moderate. Zhou et al. compared the use of SMI vs. hand dynamometry as muscle mass markers in patients with gastrointestinal cancer, in the context of GLIM criteria. The authors found that such methods presented a high agreement; thus, they proposed that muscle mass assessment can be replaced by hand grip strength, according to GLIM criteria, to diagnose malnutrition in patients with gastrointestinal cancer [44]. 

Our findings support the use of L3-CT to evaluate muscle mass in cancer inpatients, in the context of GLIM criteria. As many patients with neoplasm undergo a CT for diagnostic, staging, or follow-up purposes, this method is particularly interesting and can be helpful to identify sarcopenia and malnutrition. Thus, it would allow for early dietary, physical exercise, and, if needed, pharmacological interventions. 

Strengths of this study include the substantial number of subjects and the long-term monitoring. All measurements were performed by a single blinded reader, using a CT scan that has been routinely used for diagnostic purposes. 

All the same, this study has some limitations. It was designed as a single-center observational study, which prevents us from extracting causal conclusions. The inclusion of patients with various types of tumors and the performance of CT during the 3 months prior to hospital admission may have contributed to the heterogeneity of the study population. Finally, the sample is composed of hospitalized patients with an advanced disease, which is associated with higher mortality. However, this fact could reinforce the value of CT, since it is capable of discriminating the presence of sarcopenia even among high-risk patients.

In conclusion, low muscle mass, assessed by L3-CT, was determined to affect more than half of cancer inpatients. The deceased patients at 6 months had a lower skeletal muscle cross-sectional area and SMI. Malnutrition according to GLIM criteria using CT-determined sarcopenia was shown to adequately predict 6-month mortality. 

## Figures and Tables

**Figure 1 nutrients-13-02647-f001:**
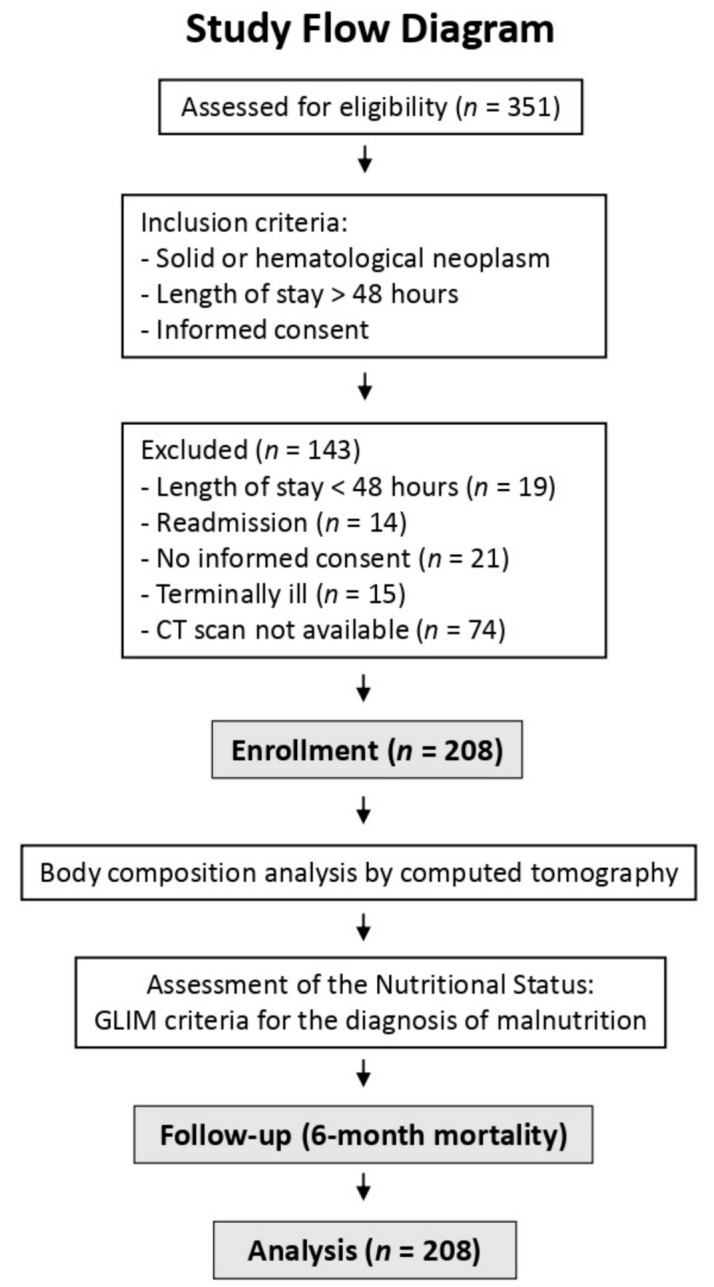
Study flow diagram and research methodology. GLIM: Global Leadership Initiative on Malnutrition.

**Figure 2 nutrients-13-02647-f002:**
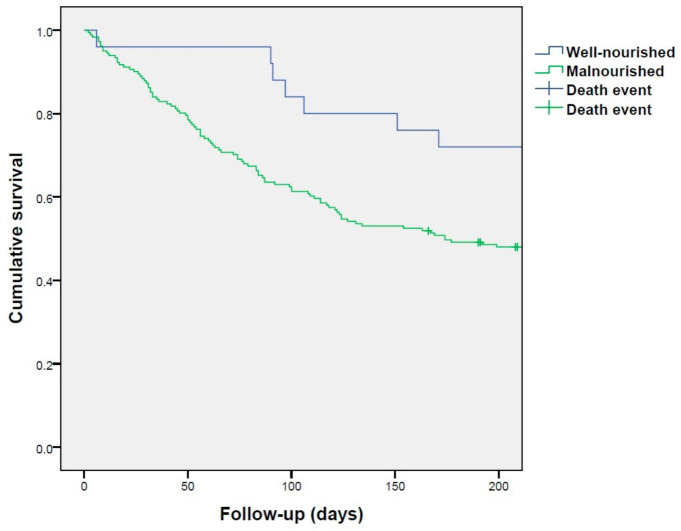
Kaplan–Meier survival analysis for subjects with malnutrition according to the GLIM criteria.

**Table 1 nutrients-13-02647-t001:** General characteristics.

		***n* = 208**
Age (years)	mean ± SD	60.5 ± 12.9
Sex	*n* (%)	
Men		115 (55.3)
Women		93 (44.7)
Type of hospital admission	*n* (%)	
Programmed		32 (15.4)
Urgent		176 (84.6)
Stage	*n* (%)	
I		4 (2)
II		9 (4.3)
III		43 (20.7)
IV		154 (74)
Glasgow prognostic score	*n* (%)	
No inflammation		9 (4.3)
Inflammation		199 (95.7)
BMI (kg/m^2^)	mean ± SD	
Men		24.9 ± 5.1
Women		24.3 ± 5.1
<cut-off points	*n* (%)	42 (20.2)
6-Month exitus	*n* (%)	104 (50)

SD: standard deviation; BMI: body mass index.

**Table 2 nutrients-13-02647-t002:** Body composition among sex groups.

	*n* = 208	Men	Women	*p* Value
Skeletal muscle cross-sectional area (cm^2^)	*M* ± *SD*	136.17 ± 32.55	98.05 ± 21.22	<0.001
Muscle attenuation (HU)	*M* ± *SD*	37.96 ± 23.99	34.71 ± 19.66	0.29
Skeletal muscle index (cm^2^/m^2^)	*M* ± *SD*	47.39 ± 12.31	38.66 ± 8.26	<0.001
Subcutaneous adipose tissue area (cm^2^)	*M* ± *SD*	131.82 ± 69.38	179.58 ± 93.78	<0.001
Visceral adipose tissue area (cm^2^)	*M* ± *SD*	178.34 ± 125.06	113.04. ± 88.51	<0.001
Subcutaneous and visceral adipose tissue attenuation (HU)	*M* ± *SD*	−82.74 ± 17.62	−84.67 ± 15.94	0.42
Fat free mass by anthropometry (kg)	*M* ± *SD*	51.05 ± 7.53	40.72 ± 6.71	<0.001
Fat free mass index by anthropometry (kg/m^2^)	*M* ± *SD*	17.73 ± 2.46	15.96 ± 2.43	<0.001
Fat free mass by BIA (kg)	*M* ± *SD*	60.77 ± 8.73	46.42 ± 6.74	<0.001
Fat free mass index by BIA (kg/m^2^)	*M* ± *SD*	21.07 ± 2.59	18.12 ± 2.22	<0.001

*SD*: standard deviation; HU: Hounsfield units; BIA: bioelectrical impedance analysis.

**Table 3 nutrients-13-02647-t003:** Pearson correlations between muscle mass, body composition, and physical function.

	BMI	Skeletal Muscle Cross-Sectional Area	Fat Free Mass (Anthropometry)	Skeletal Muscle Index	FFMI (BIA)	FFMI (Anthropometry)	Hand Grip Strength
BMI	-	0.291 a	0.558 a	0.379 a	0.784 a	0.789 a	0.154 b
Skeletal muscle cross-sectional area	0.291 a	-	0.651 a	0.897 a	0.614 a	0.467 a	0.608 a
Fat free mass (anthropometry)	0.558 a	0.651 a	-	0.458 a	0.698 a	0.840 a	0.557 a
SMI	0.379 a	0.897 a	0.458 a	-	0.579 a	0.485 a	0.452 a
FFMI (BIA)	0.784 a	0.614 a	0.698 a	0.579 a	-	0.781 a	0.481 a
FFMI (anthropometry)	0.789 a	0.467 a	0.840 a	0.485 a	0.781 a	-	0.306 a
Hand grip strength	0.154 b	0.608 a	0.557 a	0.452 a	0.481 a	0.306 a	-

BMI: body mass index; BIA: bioelectrical impedance analysis; FFMI: fat free mass index; SMI: skeletal muscle index. a = *p* < 0.01; b = *p* < 0.05.

**Table 4 nutrients-13-02647-t004:** Body composition determined by CT. Differences between survivors and deceased at 6 months.

	Survivors (*n* = 104) Mean ± SD	Deceased (*n* = 104) Mean ± SD	*p* Value
Skeletal muscle cross-sectional area (cm^2^)	126.08 ± 37.75	112.88 ± 27.89	0.003
Muscle attenuation (HU)	40.74 ± 27.48	32.32 ± 14.16	0.007
Skeletal muscle index (cm^2^/m^2^)	45.69 ± 12.94	41.28 ± 9.45	0.006
Subcutaneous adipose tissue area (cm^2^)	155.86 ± 87.69	150.67 ± 81.44	0.660
Visceral adipose tissue area (cm^2^)	153.46 ± 115.62	144.49 ± 113.98	0.575
Subcutaneous and visceral adipose tissue attenuation (HU)	−86.19 ± 17.02	−81.06 ± 16.43	0.031

SD: standard deviation; HU: Hounsfield units.

## Data Availability

The data that support the findings of this study are available from the corresponding author upon reasonable request.

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
