# Peer review of "CT-Determined Sarcopenia in GLIM-Defined Malnutrition and Prediction of 6-Month Mortality in Cancer Inpatients"

_nutrients, 2021, doi:10.3390/nu13082647_

Round 1

Reviewer 1 Report

It is a study using third lumbar vertebra-computer tomography (CT)-determined sarcopenia in malnutrition and prediction of 6-month mortality in cancer inpatients. I recommend for publication after the following points are addressed.

  • Line 24, what’s the full name for GLIM?
  • line 29, cm2/ m2 should be changed to cm2/m2. Please check all.
  • The authors should include other methods to detect the body composition, and have a comparasion with the method in this study.
  • The novelty of this study is not so clear. The introduction should be enhanced much more related to the novelty.
  • A scheme is encourage to added for illustrating the research methodology.

Author Response

Dear Reviewer,

Thank you for giving us the opportunity to improve our article “CT-determined Sarcopenia in GLIM-defined Malnutrition and Prediction of 6-month Mortality in Cancer Inpatients”.

The various suggestions have been incorporated into the new version wherever applicable. Please find below our responses and the action taken to all the suggestions and comments.

Please see the attachment to check the new version of the manuscript.

Once again, we very much appreciate all the work with the review.

Yours sincerely,

Dr. Francisco José Sánchez Torralvo

Dr. Ignacio Ruiz García

Dr. Gabriel Olveira

Comments and Suggestions for Authors

It is a study using third lumbar vertebra-computer tomography (CT)-determined sarcopenia in malnutrition and prediction of 6-month mortality in cancer inpatients. I recommend for publication after the following points are addressed.

Reviewer (R): Line 24, what’s the full name for GLIM?

Authors (A): Thank you for your appreciation. We have revised all the abbreviations.

R: line 29, cm2/ m2 should be changed to cm2/m2. Please check all.

A: Thank you for your appreciation. We have checked all the units.

R: The authors should include other methods to detect the body composition, and have a comparasion with the method in this study.

A: Thank you for your suggestion that improves our manuscript. We have added two more methods to assess the body composition: anthropometry and bioelectrical impedance analysis. 

R: The novelty of this study is not so clear. The introduction should be enhanced much more related to the novelty.

A: Thank you for your suggestion. In the new version of the manuscript, we tried to emphasize that to our knowledge, this is the first study to evaluate the use of CT as a determinant of muscle mass in application of the GLIM criteria and to establish its association with 6-month mortality in hospitalized cancer patients.

R: A scheme is encourage to added for illustrating the research methodology.

A: We very much appreciate your suggestion. We have improved the study flow diagram to include data from the research methodology.

Reviewer 2 Report

The authors reported on a series of cancer inpatients in which they evaluated the body composition by L3 CT. They found that sarcopenia is a significant predictor of 6-months mortality.

Overall, the topic is of interest, and the manuscript is well-written.

I have some comments:

1 - the authors used a multivariate logistic regression model to estimate the risk of mortality. Though this method could provide trustworthy results in specific settings, a time-dependent Cox model is typically the best choice for survival data;

2 - please, always provide raw numbers, not only percentages;

3 – the authors acknowledge the presence of various types of tumours as a limitation. It would be interesting to report the rate of patients with sarcopenia according to tumour type and to investigate the effect of sarcopenia in patients with different tumours.

4 – in the multivariate model, sex was not considered as confounder; however, in some types of tumours it is one of the main prognostic factors. Please account for the variation due to sex, or perform a subgroup analysis to report the effect of sarcopenia according to sex.

Author Response

Dear Reviewer,

Thank you for giving us the opportunity to improve our article “CT-determined Sarcopenia in GLIM-defined Malnutrition and Prediction of 6-month Mortality in Cancer Inpatients”.

The various suggestions have been incorporated into the new version wherever applicable. Please find below our responses and the action taken to all the suggestions and comments.

Please see the attachment to check the new version of the manuscript.

Once again, we very much appreciate all the work with the review.

Yours sincerely,

Dr. Francisco José Sánchez Torralvo

Dr. Ignacio Ruiz García

Dr. Gabriel Olveira

Reviewer (R):

Comments and Suggestions for Authors

The authors reported on a series of cancer inpatients in which they evaluated the body composition by L3 CT. They found that sarcopenia is a significant predictor of 6-months mortality.

Overall, the topic is of interest, and the manuscript is well-written.

R: I have some comments:

1 - the authors used a multivariate logistic regression model to estimate the risk of mortality. Though this method could provide trustworthy results in specific settings, a time-dependent Cox model is typically the best choice for survival data;

Authors (A): We very much appreciate your suggestion. We have changed the logistic regression model for a Cox model to analyze the survival data in the new version of the manuscript.

R: 2 - please, always provide raw numbers, not only percentages.

A: Thank you for your appreciation. We have provided the missing raw numbers in the new version of the manuscript.

R: 3 – the authors acknowledge the presence of various types of tumors as a limitation. It would be interesting to report the rate of patients with sarcopenia according to tumor type and to investigate the effect of sarcopenia in patients with different tumors.

A: Thank you for your suggestion. Due to the large number of tumor types included, we have considered that the sample size is small to investigate the effect of sarcopenia for each one. The prevalence of sarcopenia for each group may have been affected by the small number of cases in some groups, which makes their enumeration less interesting.

R: 4 – in the multivariate model, sex was not considered as confounder; however, in some types of tumours it is one of the main prognostic factors. Please account for the variation due to sex, or perform a subgroup analysis to report the effect of sarcopenia according to sex.

A: Thank you for your appreciation. It is actually a typographical error. The multivariate model (now Cox model) is adjusted for age, tumor stage, and sex.

Reviewer 3 Report

Overall it's a strong study however, the writing is quite disjointed. I would highly recommend that the authors go back through and make the manuscript flow better. Below is my detailed feedback

Introduction

  1. Please use patient first language and state things such as "patients with cancer" rather than "cancer patients"
  2. Please define GLIM prior to using the abbreviation
  3. Line 68-70 is confusing. Please re-write it. I've read it several times and can't understand whether you think that a CT is a convenient way to assess body composition or whether the radiation makes it inconvenient. 
  4. Line 84-86- You state some studies but don't cite any. Please cite the relevant studies.
  5. Same with the next sentence in line 86
  6. Line 89, you should state that you are unaware of any studies instead of definitively stating that there are no studies out there.
  7. You should state your objective prior to your hypothesis.

Methodology

  1. You re-stated your inclusion criteria in your exclusion criteria
  2. You should move the anthropometric measurements out of the nutritional status. Perhaps a subsection for anthropometric measurements. I understand that it's a part of the GLIM criteria, but there are 4 other criteria. You can perhaps work that in with section 2.3. I just think that the organization needs to be better. 
  3. In the data analysis section, wherever you use non-parametric tests please provide the median instead of the mean. However, you may want to investigate large sample theory (Chernoff, 1956, Lehman, 2004), which might be able to provide you with justification to run parametric analyses on all of your data.

Results

  1. Instead of writing NS provide specific p-value in Table 2
  2. Where are the results of the Chi-square and Fishers test?

Discussion

  1. In the first paragraph of the discussion please re-state the objective and how your study addressed this objective. The first sentence is rather confusing as it currently reads. I had to go back and re-read your results to confirm.
  2. Cite after Huang, et al, on line 195
  3. I am having a hard time trying to get a takeaway message from this manuscript as the discussion currently seems rather disjointed. I would definitely recommend improving the flow of the discussion.

Author Response

Dear Reviewer,

Thank you for giving us the opportunity to improve our article “CT-determined Sarcopenia in GLIM-defined Malnutrition and Prediction of 6-month Mortality in Cancer Inpatients”.

The various suggestions have been incorporated into the new version wherever applicable. Please find below our responses and the action taken to all the suggestions and comments.

Please see the attachment to check the new version of the manuscript.

Once again, we very much appreciate all the work with the review.

Yours sincerely,

Dr. Francisco José Sánchez Torralvo

Dr. Ignacio Ruiz García

Dr. Gabriel Olveira

Reviewer:

Overall it's a strong study however, the writing is quite disjointed. I would highly recommend that the authors go back through and make the manuscript flow better. Below is my detailed feedback

Introduction

  1. Please use patient first language and state things such as "patients with cancer" rather than "cancer patients"

Authors (A): Thank you for your appreciation. We have prioritized the use of patient-first language when possible.

  1. Please define GLIM prior to using the abbreviation

A: Thank you for your appreciation. We have revised all the abbreviations.

  1. Line 68-70 is confusing. Please re-write it. I've read it several times and can't understand whether you think that a CT is a convenient way to assess body composition or whether the radiation makes it inconvenient. 

A: Thank you for your appreciation that improves our manuscript. We have erased the wrong statement.

  1. Line 84-86- You state some studies but don't cite any. Please cite the relevant studies.

A: We very much appreciate your suggestion. We have reviewed the bibliographic references in depth and we have solved the errors detected.

  1. Same with the next sentence in line 86

A: Thank you for your appreciation. We have changed the sentence.

  1. Line 89, you should state that you are unaware of any studies instead of definitively stating that there are no studies out there.

A: Thank you for your appreciation. We have changed the sentence.

  1. You should state your objective prior to your hypothesis.

A: Thank you for your appreciation.  We have changed the order.

Methodology

  1. You re-stated your inclusion criteria in your exclusion criteria

A: Thank you for your appreciation. We have deleted the re-stated criteria.

  1. You should move the anthropometric measurements out of the nutritional status. Perhaps a subsection for anthropometric measurements. I understand that it's a part of the GLIM criteria, but there are 4 other criteria. You can perhaps work that in with section 2.3. I just think that the organization needs to be better. 

A: We very much appreciate your suggestion. We have unified the two sections to achieve a more coherent organization.

  1. In the data analysis section, wherever you use non-parametric tests please provide the median instead of the mean. However, you may want to investigate large sample theory (Chernoff, 1956, Lehman, 2004), which might be able to provide you with justification to run parametric analyses on all of your data.

A: Thank you for your appreciation that improves our manuscript. We have revised the data analysis section and consulted with our statisticians and, indeed, we run parametric analyses according to large sample theory (our sample size is greater than 30). We have changed the data analysis section accordingly.

Results

  1. Instead of writing NS provide specific p-value in Table 2

A: Thank you for your appreciation. We have provided the specific p-values in the new version.

  1. Where are the results of the Chi-square and Fishers test?

A: Thanks for your appreciation. The Chi-square test was used to verify the association between the diagnosis of malnutrition and mortality at 6 months, although the data represented in the text correspond to the results of the final logistic regression.

On the other hand, there is an error in the sentence. Indeed, the use of Fisher's test was not necessary for this statistical analysis. We have removed that statement.

Discussion

  1. In the first paragraph of the discussion please re-state the objective and how your study addressed this objective. The first sentence is rather confusing as it currently reads. I had to go back and re-read your results to confirm.

A: Thank you for your appreciation that improves our manuscript. We have simplified the sentence in the new version for a better understanding.

  1. Cite after Huang, et al, on line 195

A: Thank you for your suggestion. We have added the cite.

  1. I am having a hard time trying to get a takeaway message from this manuscript as the discussion currently seems rather disjointed. I would definitely recommend improving the flow of the discussion.

A: Thank you for your appreciation that improves our manuscript. We have tried to improve the flow of the discussion in the new version of the manuscript.

Round 2

Reviewer 1 Report

The revised version is suggested to be published.

Author Response

Dear Reviewer,

Once again, we very much appreciate all the work with the review.

Yours sincerely,

Dr. Francisco José Sánchez Torralvo

Dr. Ignacio Ruiz García

Dr. Gabriel Olveira

Reviewer 2 Report

The authors have satisfactorily answered to all my comments.

Reviewer 3 Report

Thank you for taking the time to respond to all of my queries. Below are a few more minor suggestions.

line 53- patients with cancer

line 60- please define GLIM (I know you define it in your abstract, but you should also define it in your introduction prior to using it in the rest of your manuscript)

lines 88-90- you state that studies are scarce. Does this mean that none exist or a few exist? I'm assuming based on the fact that you haven't cited anything that you mean to say that none exist. A better way to say this might be "The authors are unaware of any studies that use CT to assess...."

line 93-94- do you mean patients with cancer? Inpatient patients with cancer?

Methods

I appreciate the authors taking the route of using Large Sample Theory to use parametric analyses. I think the authors should still write a sentence in the statistical analysis section that states how they evaluated for normality and the fact that they chose to use parametric analyses due to large sample theory (cite the authors of large sample theory).

Author Response

Dear Reviewer,

Thank you for giving us the opportunity to improve our article “CT-determined Sarcopenia in GLIM-defined Malnutrition and Prediction of 6-month Mortality in Cancer Inpatients”.

The various suggestions have been incorporated into the new version wherever applicable. Please find below our responses and the action taken to all the suggestions and comments.

Please see the attachment to check the new version of the manuscript.

Once again, we very much appreciate all the work with the review.

Yours sincerely,

Dr. Francisco José Sánchez Torralvo

Dr. Ignacio Ruiz García

Dr. Gabriel Olveira

Reviewer (R)

Thank you for taking the time to respond to all of my queries. Below are a few more minor suggestions.

line 53- patients with cancer

Authors (A):  Thank you for your appreciation. We have corrected the expression

R: line 60- please define GLIM (I know you define it in your abstract, but you should also define it in your introduction prior to using it in the rest of your manuscript)

A: Thank you for your appreciation. We have added the definition

R: lines 88-90- you state that studies are scarce. Does this mean that none exist or a few exist? I'm assuming based on the fact that you haven't cited anything that you mean to say that none exist. A better way to say this might be "The authors are unaware of any studies that use CT to assess...."

A: Thank you for your appreciation. We refer to the studies cited with references 32 y 33 (up to our knowledge, there are only two studies assessing the use of CT to measure muscle mass in application of GLIM criteria as a marker of adverse clinical outcomes).

R: line 93-94- do you mean patients with cancer? Inpatient patients with cancer?

A: Thank you for your appreciation. We have changed the expression to "hospitalized patients with cancer" for a better understanding.

R: Methods

I appreciate the authors taking the route of using Large Sample Theory to use parametric analyses. I think the authors should still write a sentence in the statistical analysis section that states how they evaluated for normality and the fact that they chose to use parametric analyses due to large sample theory (cite the authors of large sample theory).
A:  We very much appreciate your suggestion. We have added a related sentence and cites in the statistical analysis section.